# Virescenosides from the Holothurian-Associated Fungus *Acremonium striatisporum* Kmm 4401

**DOI:** 10.3390/md17110616

**Published:** 2019-10-29

**Authors:** Olesya I. Zhuravleva, Alexandr S. Antonov, Galina K. Oleinikova, Yuliya V. Khudyakova, Roman S. Popov, Vladimir A. Denisenko, Evgeny A. Pislyagin, Ekaterina A. Chingizova, Shamil Sh. Afiyatullov

**Affiliations:** 1G.B. Elyakov Pacific Institute of Bioorganic Chemistry, Far Eastern Branch of the Russian Academy of Sciences, Prospect 100-letiya Vladivostoka, 159, Vladivostok 690022, Russia; zhuravleva.oi@dvfu.ru (O.I.Z.); pibocfebras@gmail.com (A.S.A.); oleingk@mail.ru (G.K.O.); 161070@rambler.ru (Y.V.K.); prs_90@mail.ru (R.S.P.); vladenis@pidoc.dvo.ru (V.A.D.); pislyagin@hotmail.com (E.A.P.); martyyas@mail.ru (E.A.C.); 2School of Natural Science, Far Eastern Federal University, Sukhanova St., 8, Vladivostok 690000, Russia

**Keywords:** *Acremonium striatisporum*, secondary metabolites, marine fungi, diterpene glycosides, urease activity

## Abstract

Ten new diterpene glycosides virescenosides Z_9_-Z_18_ (**1**–**10**) together with three known analogues (**11**–**13**) and aglycon of virescenoside A (**14**) were isolated from the marine-derived fungus *Acremonium striatisporum* KMM 4401. These compounds were obtained by cultivating fungus on wort agar medium with the addition of potassium bromide. Structures of the isolated metabolites were established based on spectroscopic methods. The effects of some isolated glycosides and aglycons **15**–**18** on urease activity and regulation of Reactive Oxygen Species (ROS) and Nitric Oxide (NO) production in macrophages stimulated with lipopolysaccharide (LPC) were evaluated.

## 1. Introduction

Marine fungi are promising and prolific sources of new biologically active compounds. At the same time, glycosylated secondary metabolites of marine fungi such as ribofuranosides, containing as aglycon moieties anthraquinones [1,2,3], diphenyl ethers [4,5], isocoumarin [6] and naphthyl derivatives [7] are relatively rare. Recently, two steroid glycosides with β-D-mannose as sugar part were isolated from ascomycete *Dichotomomyces cejpii* [8] and new triterpene glycoside auxarthonoside bearing rare sugar N-acetyl-6-methoxy-glucosamine was described from sponge-derived fungus *Auxarthron reticulatum* [9]. Some of these glycosides exhibited cytotoxic [5], radical scavenging [3,4], and neurotropic [8] activities.

During our ongoing search for new natural compounds from marine-derived fungi, we have investigated the strain *Acremonium striatisporum* KMM 4401 associated with the holothurian *Eupentacta fraudatrix*. Twenty-one new diterpene glycosides, virescenosides have previously been isolated from this strain under cultivation on solid rice medium and wort agar medium [10,11,12]. Virescenosides Z_5_ and Z_7_ exhibited an unusual 16-chloro-15-hydroxyethyl group as their side chains in aglycones [12]. So, we attempted directed biosynthesis for the production of other halogenated compounds by culturing the fungus *Acremonium striatisporum* KMM 4401 in media containing potassium bromide. Unfortunately, we were unable to obtain glycoside derivatives with the incorporation of a bromine atom in a molecule structure. Chromatographic separation of the CHCl_3_-EtOH extract of the culture of fungus has now led to the isolation of ten undescribed diterpene glycosides virescenosides Z_9_-Z_18_ (**1**‒**10**) (Figure 1) together with known virescenosides F (**11**) and G (**12**), lactone of virescenoside G (**13**) and aglycon of virescenoside A (**14**) (Appendix A).

## 2. Results and Discussion

The CHCl_3_-EtOH (2:1, v/v) extract of the culture of *A. striatisporum* was separated by low-pressure reversed-phase column chromatography on Teflon powder Polycrome-1 followed by Si gel flash column chromatography and then by RP HPLC to yield individual compounds **1**‒**14** as colorless, amorphous solids.

The molecular formula of virescenoside Z_9_ (**1**) was determined as C_26_H_42_O_11_ based on the analysis of HRESIMS (*m/z* 529.2656 [M–H]^−^, calcd for C_26_H_41_O_11_, 529.2654) and NMR data. A close inspection of the ^1^H and ^13^C NMR data (Table 1 and Table 2; Appendix A) of **1** by DEPT and HSQC revealed the presence of three quaternary methyls (δ_H_ 0,95, 1.28, 1.81; δ_C_ 28.5, 17.7, 25.8), six methylenes (δ_C_ 18.4, 34.3, 46.9, 49.8, 64.0 and 74.0), including two oxygen-bearing, eight oxygenated methines (δ_H_ 3.61, 3.70, 4.28, 4.50, 4.56, 4.69, 4.74, 5.43; δ_C_ 84.7, 57.3, 69.1, 75.7, 72.7, 67.2, 72.3, 101.2) including one methine linked to an anomeric carbon, two tertiary (δ_H_ 1.93, 2.41; δ_C_ 60.5, 55.9), four saturated quaternary carbons (δ_C_ 35.8, 43.8 (2C) and 80.2), including one oxygen-bearing, one monosubstituted double bond (δ_C_ 108.4, 151.0) and one carbonyl or carboxyl carbon (δ_C_ 178.0). HMBC correlations from H_3_-20 (δ_H_ 1.28) to C-1 (δ_C_ 46.9), C-5 (δ_C_ 55.9), C-9 (δ_C_ 60.5) and C-10 (δ_C_ 43.8), from H_3_-18 (δ_H_ 1.81) to C-3 (δ_C_ 84.7), C-4 (δ_C_ 43.8), C-5 (δ_C_ 55.9) and C-19 (δ_C_ 74.0), from H-3 (δ_H_ 3.61) to C-2 (δ_C_ 69.1), C-4 and C-19, from H-1β (δ_H_ 2.34) to C-3, from H-6 (δ_H_ 3.70) to C-4, C-5, C-7 (δ_C_ 178.0) and C-8 (δ_C_ 80.2), from H-9 (δ_H_ 1.93) to C-8 and C-10 established the structures of the A and B rings and the location of hydroxy groups at C-2, C-3, C-6, C-8 and carbonyl function at C-7. The correlations observed in the COSY and HSQC spectra of **1** indicated the presence of the isolated spin system: >CH‒CH_2_‒CH_2_‒ (C-9‒C-11‒C-12). These data and HMBC correlations from H_3_-17 (δ_H_ 0.95) to C-12 (δ_C_ 34.3), C-13 (δ_C_ 35.8), C-14 (δ_C_ 49.8), C-15 (δ_C_ 151.0) and from H-14β (δ_H_ 1.48) to C-8, C-9 and C-12 established the structure of the C ring in **1**.

The proton signals of a typical ABX system of a vinyl group at δ_H_ 6.64 (1H, dd, 10.8, 17.6 Hz), 4.96 (1H, dd, 1.8, 17.6) and 4.85 (1H, dd, 1.8, 10.8) indicated the C-15, C-16 position of this double bond [13,14,15,16]. NOE correlations (Figure 2) H_3_-20 (δ_H_ 1.28)/H-2 (δ_H_ 4.28), H-6 (δ_H_ 3.70), H-19b (δ_H_ 4.96) and H-5 (δ_H_ 2.41)/H-3 (δ_H_ 3.62), H_3_-18 (δ_H_ 1.81) indicated a *trans*-ring fusion of the A and B rings, as well as the stereochemistry of the methyl and hydroxymethyl groups at C-4, methyl group at C-10 and hydroxy groups at C-2, C-3 and C-6. NOE cross-peaks H-9 (δ_H_ 1.93)/H-5 and H-14β (δ_H_ 1.48)/H_3_-20, H_3_-17 (δ_H_ 0.95), H-6 showed the stereochemistry of the methyl group at C-13 and suggested the β-orientation of hydroxy group at C-8.

Interpretation of the COSY data gave rise to spin systems for monosaccharide involving one anomeric proton, four oxymethines and protons of a hydroxymethyl group. A comparison of the ^13^C NMR spectrum of **1** with the data published for α-D-altropyranoses and β-D-altropyranoses as well as a good coincidence of carbon signals due to the glycosidic moiety with those of virescenosides O, T, W [10] together with magnitudes of ^1^H-^1^H spin coupling constants in ^1^H NMR spectra of **1** elucidated the presence of a β-D-altropyranoside unit of ^4^C_1_ form in **1**. A long-range correlation H-1′ (δ_H_ 5.43)/C-19 as well as the NOESY cross-peak between H-1′ and H-19a and downfield chemical shift of C-19 (δ_C_ 74.0) revealed a linkage between the altrose and aglycon. Thus, the structure of virescenoside Z_9_ (**1**) was represented as 19-*O*-β-D-altropyranosyl-7-oxo-isopimara-15-en-2α,3β,6α,8β-tetraol.

In HRESIMS virescenoside Z_10_ (**2**) gave a quasimolecular ion at *m/z* 493.2446 [M–H]^−^. These data, coupled with ^13^C NMR spectral data (DEPT), established the molecular formula of **2** as C_26_H_38_O_9_. ^1^H and ^13^C NMR spectra of **2** (Table 1 and Table 2; Appendix A) indicated the presence of a ∆^15^-pimarene-type aglycon possessing primary alcohol on a quaternary carbon (AB system, coupling at 3.73 d, 10.2 Hz and 4.17 d, 10.2 Hz) and one secondary alcohol function at δ_C_ 80.0. The remaining functionality, corresponding to the carbon signals at δ 202.9 (C), 168.7 (C) and 130.3 (C), suggested the presence of the tetrasubstituted enone chromophore. The structure of the aglycon part of **2** was found by extensive NMR spectroscopy to be the same as that of virescenoside P [17].

The HRESIMS of virescenosides Z_11_ (**3**) showed the quasimolecular ion at *m/z* 509.2408 [M–H]^−^. These data, coupled with ^13^C NMR spectral data (DEPT), established the molecular formula of **3** as C_26_H_38_O_10_. The structure of the aglycon moiety of **3** was found by extensive NMR spectroscopy (^1^H, ^13^C, HSQC, HMBC and NOESY) (Table 1 and Table 2; Appendix A) to be the same as those of virescenoside M [18].

The ^13^C and ^1^H NMR spectra of the sugar moieties of virescenoside Z_10_ (**2**) and Z_11_ (**3**) showed a close similarity of all proton and carbon chemical shifts with those of virescenosides Z_7_ and Z_8_ [12]. The 7.7-,7.4-Hz splitting between H-2 and H-3 indicated that both were axial, whereas the 4.8-, 5.7-Hz splitting between H-4 and H-5 showed that these protons in equatorial position. These data and HMBC correlations between anomeric protons and C-5′-methine groups and between H-5′ (δ_H_ 4.24, 4.22) and C-6′ (δ_C_ 174.0) suggested the presence of a *β*-altruronopyranoside unit of ^1^C_4_ conformation in **2** and **3**. The long-range correlations H-1′ (δ_H_ 4.84, 4.82)/C-19 as well as the NOESY cross-peak between H-1′ and H-19a and downfield chemical shifts of C-19 (δ_C_ 73.3, 73.8) indicated that sugar moieties were linked at C-19. Earlier in result of reduction of the sum of virescenosides Z_4_‒Z_8_ with LiAlH_4_ and the acid hydrolysis of obtained products was isolated D-altrose as the only sugar that was identified by GLC of the corresponding acetylated (+)- and (‒)-2-octyl glycosides using authentic samples prepared from D-altrose [12]. Thus, the structure of virescenoside Z_10_ (**2**) was determined as 19-*O*-β-D-altruronopyranosyl-7-oxo-isopimara-8(9),15-dien-3β-ol, and the structure of virescenoside Z_11_ (**3**) was established as 19-*O*-β-D-altruronopyranosyl-7-oxo-isopimara-8(9),15-dien-2α,3β-diol.

The HRESIMS of virescenosides Z_12_ (**4**) and Z_13_ (**5**) showed the quasimolecular ions at *m/z* 517.2770 [M + Na]^+^ and *m/z* 533.2718 [M + Na]^+^, respectively. These data, coupled with ^13^C NMR spectral data (DEPT), established the molecular formula of **4** and **5** as C_27_H_42_O_8_ and C_27_H_42_O_9_, respectively. A close inspection of the ^1^H and ^13^C NMR data of **4** (Table 1 and Table 2; Appendix A) revealed that virescenoside Z_12_ (**4**) was structurally identical to virescenosides B [13] and G [19] (See Extraction and Isolation) with respect to the aglycon. The structure of the aglycon moiety of **5** was found by extensive NMR spectroscopy (Table 1 and Table 2; Appendix A) to be the same as that of virescenosides A [13,20] and F [19] (See Extraction and Isolation).

The NMR spectra of glycosides **4** and **5** indicated that both compounds contained closed carbohydrate moieties (Table 1 and Table 2). Initial examination of the 1-D proton and one bond correlation NMR data suggested the presence of one sugar (anomeric signals at δ_H_ 4.85, δ_C_ 103.7 for **4** and δ_H_ 4.85, δ_C_ 103.3 for **5**). The ^1^H and ^13^C NMR spectra of the sugar parts of **4** and **5** indicated the presence of the methoxy groups (both, δ_H_ 3.78, δ_C_ 53.3). HMBC correlations from anomeric protons to C5′-methine groups and from H-5′ (δ_H_ 4.28) to C-6′ (δ_C_ 172.7, 172.9) and from H_3_-7′ (δ_H_ 3.78) to C-6′ together with magnitudes of ^1^H-^1^H spin coupling constants suggested the presence of the methyl ester of a *β*-altruronopyranoside unit of ^1^C_4_ form in **4** and **5**. A long-range correlations H-1′ (δ_H_ 4.85)/C-19 (δ_C_ 73.9, 74.1) as well as the NOESY cross-peaks between H-1′ and H-19a (δ_H_ 3.83, 3.72) and downfield chemical shifts of C-19 indicated that sugar moieties were linked at C-19. Thus, the structure of virescenoside Z_12_ (**4**) was determined as 19-*O*-[(methyl-β-D-altruronopyranosyl)-uronat]-isopimara-7,15-dien-3β-ol, and the structure of virescenoside Z_13_ (**5**) was established as 19-*O*-[(methyl-β-D-altruronopyranosyl)-uronat]-isopimara-7,15-dien-2α,3β-diol.

The NMR data (Table 1 and Table 3) of virescenosides Z_14_ (**6**), Z_15_ (**7**) and Z_16_ (**8**) suggested the presence of one sugar (anomeric signals at δ_H_ 4.78, δ_C_ 102.8, δ_H_ 4.79, δ_C_ 102.8, δ_H_ 4.75, δ_C_ 101.9). The ^1^H and ^13^C NMR spectra of the sugar moieties of **6**, **7** and **8** showed a close similarity of all proton and carbon chemical shifts and proton multiplicities. These data and HMBC correlations from anomeric protons to C-5′ methine groups and from H-5′ (δ_H_ 4.24, 4.24, 4.23) to C-6′ (δ_C_ 173.2, 172.8, 172.6) and from H_3_-7′ (δ_H_ 3.76, 3.76, 3.77) to C-6′ suggested the presence of the methyl ester of a *β*-altruronopyranoside unit in **6**, **7** and **8**. The 7.0-, 7.3-, 8.0-Hz splitting between H-4 and H-5 indicated that both were axial and conformation of sugar parts in **6**, **7** and **8** is ^4^C_1_.

The HRESIMS of virescenosides Z_14_ (**6**) showed the quasimolecular ion at *m/z* 547.2508 [M + Na]^+^. These data, coupled with ^13^C NMR spectral data (DEPT), established the molecular formula of **6** as C_27_H_40_O_10_. The structure of the aglycon moiety of **6** was found by extensive NMR spectroscopy (^1^H, ^13^C, HSQC, HMBC and NOESY) (Table 1 and Table 3; Appendix A) to be the same as those of virescenoside V [21].

The molecular formula of virescenoside Z_15_ (**7**) was determined as C_27_H_40_O_10_ based on the analysis of HRESIMS (*m/z* 523.2550 [M-H]^−^, calcd for C_27_H_39_O_10_, 523.2549) and NMR data. The ^1^H and ^13^C NMR data (Table 1 and Table 3; Appendix A) observed for the aglycon part of **7** closely resembled those obtained for virescenoside Z_10_ (**2**) with the exception of the C-1‒C-4 carbon and proton signals of ring A. The HMBC correlations from H-5 (δ_H_ 1.75) to C-3 (δ_C_ 84.4), H-3 (δ_H_ 3.00) and from H_2_-1 (δ_H_ 1.23, 2.18) to C-2 (δ_C_ 69.2) and downfield chemical shifts of C-2 placed an additional hydroxy group at C-2 of ring A. The relative stereochemistry of protons on C-2 and C-3 was defined based on the ^1^H-^1^H coupling constant (*J*=9.8) and assigned as axial. Previously, a similar aglycon has been described for virescenoside M [10].

The HRESIMS of virescenoside Z_16_ (**8**) showed the quasimolecular at *m/z* 515.2617 [M + Na]^+^. These data, coupled with ^13^C NMR spectral data (DEPT), established the molecular formula of **8** as C_27_H_40_O_8_ (Table 1 and Table 3). The structure of the aglycon moiety of **8** was found by 2D NMR experiments (Appendix A) to be the same as that of virescenoside Z_4_ [12].

The attachment of a carbohydrate chains at C-19 of aglycon moieties of **6**, **7** and **8** was confirmed by cross-peaks H-1′ (δ_H_ 4.78, 4.79, 4.75)/H-19a (δ_H_ 3.68, 3.65, 3.90) and H-1′/C-19 (δ_C_ 73.1, 73.6, 75.1) in the NOESY and HMBC spectra, respectively. From all these data, virescenoside Z_14_ (**6**) was structurally identified as 19-*O*-[(methyl-β-D-altruronopyranosyl)-uronat]-7-oxo-isopimara-8(14),15-dien-2α,3β-diol, virescenoside Z_15_ (**7**) as 19-*O*-[(methyl-β-D-altruronopyranosyl)-uronat]-7-oxo-isopimara-8(9),15-dien-2α,3β-diol and virescenoside Z_16_ (**8**) as 19-*O*-[(methyl-β-D-altruronopyranosyl)-uronat]-3-oxo-isopimara-7,15-dien.

The HRESIMS of virescenoside Z_17_ (**9**) showed the quasimolecular ion at *m/z* 575.3194 [M + Na]^+^. These data, coupled with ^13^C NMR spectral data (DEPT), established the molecular formula of **9** as C_30_H_48_O_9_. The ^1^H and ^13^C NMR data observed for aglycon and sugar (C-1′‒C-6′) parts of **9** (Table 1 and Table 3; Appendix A) matched those reported for virescenoside Z_13_ (**5**). The correlations observed in the COSY and HSQC spectra of **9** indicated the presence of the isolated spin system: ‒CH_2_‒CH_2_‒CH_2_‒CH_3_ (C-7′‒C-10′). These data and HMBC correlations from H_3_-10′ (δ_H_ 0.96) to C-8′ (δ_C_ 32.3), C-9′ (δ_C_ 20.7) and from Ha-7′ (δ_H_ 4.15) to C-6′ (δ_C_ 172.9), C-8′ and C-9′ suggested the presence of the butyl ester of a *β*-altruronopyranoside unit of ^1^C_4_ form in **9**. On the basis of all the data above, the structure of virescenosides Z_17_ (**9**) was established as 19-*O*-[(butyl-β-D-altruronopyranosyl)-uronat]-isopimara-7,15-dien-2α,3β-diol.

The HRESIMS of virescenoside Z_18_ (**10**) showed the quasimolecular at *m/z* 517.2773 [M + Na]^+^. These data, coupled with ^13^C NMR spectral data (DEPT), established the molecular formula of **10** as C_27_H_42_O_8_. The ^1^H and ^13^C NMR data observed for the aglycon part of **10** (Table 1 and Table 3; Appendix A) matched those reported for virescenoside Q [17]. Initial examination of the 1-D proton and one bond correlation NMR data suggested the presence of one sugar (anomeric signal at δ_H_ 4.97, δ_C_ 103.5). The ^1^H and ^13^C NMR spectra of the sugar part of **10** indicated the presence of the methoxycarbonyl group (δ_H_ 3.64, δ_C_ 51.8, 170.7). A comparison of the ^13^C NMR spectrum with the data published for *α*- and *β*-D-mannopyranoses as well as a good coincidence of carbon signals C-1′‒C-4′ with those of virescenoside Q together with magnitudes of ^1^H-^1^H spin coupling constants in ^1^H NMR spectrum of **10** elucidated the presence of *β*-D-mannouronopyranoside unit of ^4^C_1_ form in **10** [17,22,23]. A long-range correlation H-1′ (δ_H_ 4.97)/C-19 (δ_C_ 72.1) as well as the NOESY cross-peak between H-1′ and H-19a (δ_H_ 4.26) and downfield chemical shifts of C-19 indicated that sugar moiety was linked at C-19. Thus, the structure of virescenoside Z_18_ (**10**) was determined as 19-*O*-[(methyl-β-D-mannopyrananosyl)-uronat]-isopimara-7,15-dien-3β-ol.

Since methanol is used in the isolation procedure of virescenosides, it is possible that the methyl esters of the sugar units may be obtained during the course of isolation. Therefore, we separated the part of subfraction II by RP-HPLC using acetonitrile instead of methanol and obtain virescenosides Z_12_ (**4**) and Z_13_ (**5**) which were characterized by ^1^H and ^13^C NMR spectra. Furthermore, we observed compounds **4**-**8** and **10** in subfraction II by HPLC-MS method (See Appendix A).

The structures of known compounds virescenosides F (**11**) and G (**12**), lactone of virescenoside G (**13**) [19] and aglycon of virescenoside A (**14**) [13] (See Appendix A) were determined based on HRESIMS and NMR data and comparison with literature. The aglycons of virescenosides B (**15**, **16**), C (**17**) and M (**18**) (See Appendix A, Experimental Section) were prepared as a result of acid hydrolysis of the corresponding glycosides for examination of their biological activity.

Next, we investigated the effects of some isolated compounds and aglycones **15**‒**18** on urease activity and regulation of ROS and NO production in macrophages stimulated with lipopolysaccharide (LPS).

The development of urease inhibitors, usually considered as antiulcer agents, carries a significant interest for medicinal chemists. Urease is an enzyme that is clinically used as diagnostic to determine the presence of pathogens in the gastrointestinal and urinary tracts. It has been described that the bacterial urease causes many clinically harmful infections, like stomach cancer, infectious stones and peptic ulcer formation in human and animal health [24]. Urease is also involved in the pathogenesis of hepatic coma, urolithiasis, urinary catheter encrustation and oral cavity infections by hydrolyzing the salivary urea [25].

Aglycons **14** and **15** inhibit urease activity with an IC_50_ of 138.8 and 125.0 *μ*M, respectively. Thiourea used as positive control inhibited urease activity with IC_50_ of 23.0 *μ*M.

Compounds **1**, **2**, **5**, **15**‒**18** at a concentration of 10 *μ*M induced a significant down-regulation of ROS production in macrophages stimulated with lipopolysaccharide (LPS) (Figure 3). Virescenoside Z_10_ (**2**) decreased the ROS content in macrophages by 45%.

Compounds **2**, **5**, **16** and **17** induced a moderate down-regulation of NO production in LPS-stimulated macrophages at concentration of 1 *μ*M (Figure 4).

## 3. Materials and Methods

### 3.1. General Experimental Procedures

Optical rotations were measured on a Perkin-Elmer 343 polarimeter (Perkin Elmer, Waltham, MA, USA). UV spectra were recorded on a Shimadzu UV-1601PC spectrometer (Shimadzu Corporation, Kyoto, Japan) in methanol. NMR spectra were recorded in CD_3_OD, CDCl_3_, DMSO-d_6_ and C_5_D_5_N on a Bruker DPX-500 (Bruker BioSpin GmbH, Rheinstetten, Germany) and a Bruker DRX-700 (Bruker BioSpin GmbH, Rheinstetten, Germany) spectrometer, using TMS as an internal standard. The Bruker Impact II Q-TOF mass spectrometer (Bruker Daltonics, Bremen, Germany) was used to record the MS and MS/MS spectra within m/z range 50–1500. The capillary voltage was set to 1300 V, and the drying gas was heated to 150 °C at the flow rate 3 L/min. Collision-induced dissociation (CID) product ion mass spectra were obtained using nitrogen as the collision gas. The instrument was operated using the program otofControl (ver. 4.0, Bruker Daltonics, Bremen, Germany) and the data were analyzed using the DataAnalysis Software (ver. 4.3, Bruker Daltonics, Bremen, Germany).

Low-pressure liquid column chromatography was performed using silica gel (50/100 μm, Imid, Russia) and Polychrome-1 (powder Teflon, Biolar, Latvia). Plates precoated with silica gel (5–17 μm, 4.5 × 6.0 cm, Imid) and silica gel 60 RP-18 F_254_S (20 × 20 cm, Merck KGaA, Germany) were used for thin-layer chromatography. Preparative HPLC was carried out on a Agilent 1100 chromatography (Agilent Technologies, USA) using a YMC ODS-AM (YMC Co., Ishikawa, Japan) (5 µm, 10 × 250 mm) and YMC ODS-A (YMC Co., Ishikawa, Japan) (5 µm, 4.6 × 250 mm) columns with a Agilent 1100 refractometer (Agilent Technologies, USA).

### 3.2. Cultivation of Fungus

The fungus was grown stationary at 22 °C for 14 days on 6 flasks (1 L) (medium: wort-200 mL, sodium tartrate-0.05 g/L, agar-20 g/L, potassium bromide-30 g/L, seawater-800 mL).

### 3.3. Extraction and Isolation

At the end of the incubation period, the mycelium and medium were homogenized and extracted three times with a mixture of CHCl_3_–EtOH (2:1, v/v, 2.5 L). The combined extracts (4.5 g) were concentrated to dryness and separated by low pressure RP CC (the column 20 × 8 cm) on Polychrome-1 Teflon powder in H_2_O and 50% EtOH. After elution of inorganic salts and highly polar compounds by H_2_O, 50% EtOH was used to obtain the fraction of amphiphilic compounds, including the virescenosides. After evaporation of the solvent, the residual material (2.6 g) was subjected to Si gel flash CC (7 × 13 cm) chromatography with a solvent gradient system of increasing polarity from 10 to 60% EtOH in CHCl_3_ (total volume 8 L). Fractions of 20 mL were collected and combined by TLC examination to obtain two subfractions. Subfraction I (CHCl_3_−EtOH 5:1, 3:1, 180 mg) was purified and separated by RP HPLC on a YMC ODS-A column eluting with MeOH−H_2_O‒TFA (85:15:0.1) to yield **8** (2.4 mg), **9** (3.6 mg), **13** (2.4 mg) and **14** (4.0 mg). Subfraction II (CHCl_3_−EtOH 2:1, 840 mg) was purified by RP HPLC on a YMC ODS-AM column eluting at first with MeOH−H_2_O‒TFA (80:20:0.1) and then with MeOH−H_2_O‒TFA (70:30:0.1) to yield **1** (2.5 mg), **2** (2.5 mg), **3** (7.5 mg), **4** (15.5 mg), **5** (71 mg), **6** (1.4 mg), **7** (6.6 mg) **10** (1.4 mg), **11** (98 mg) and **12** (63 mg).

The part of subfraction II (35 mg) was purified by RP HPLC on a YMC ODS-A column eluting with CH_3_CN−H_2_O‒TFA (50:50:0.1) to yield **4** (1.1 mg), **5** (4.5 mg), **11** (6 mg) and **12** (2 mg).

### 3.4. Spectral Data

Virescenoside Z_9_ (**1**): amorphous solids; [*α*]${}_{D}^{20}$ +1.5 (*c* 0.15, MeOH); ^1^H and ^13^C NMR data, see Table 1 and Table 2, Appendix A; HRESIMS *m*/*z* 553.2618 [M + Na]^+^ (calcd. for C_26_H_42_O_11_Na, 553.2619, Δ + 0.2 ppm).

Virescenoside Z_10_ (**2**): amorphous solids; [*α*]${}_{D}^{20}$ +10.0 (*c* 0.09, MeOH); UV (MeOH) *λ*_max_ (log *ε*) 248 (3.91) nm; ^1^H and ^13^C NMR data, see Table 1 and Table 2, Appendix A; HRESIMS *m*/*z* 493.2446 [M–H]^−^ (calcd. for C_26_H_37_O_9_, 493.2443, Δ–0.6 ppm).

Virescenoside Z_11_ (**3**): amorphous solids; [*α*]${}_{D}^{20}$ + 7.5 (*c* 0.10, MeOH); UV (MeOH) *λ*_max_ (log *ε*) 248 (3.64) nm; ^1^H and ^13^C NMR data, see Table 1 and Table 2, Appendix A; HRESIMS *m*/*z* 509.2403 [M–H]^−^ (calcd. for C_26_H_37_O_10_, 509.2392, Δ−2.0 ppm).

Virescenoside Z_12_ (**4**): amorphous solids; [*α*]${}_{D}^{20}$ −50.0 (*c* 0.10, MeOH); ^1^H and ^13^C NMR data, see Table 1 and Table 2, Appendix A; HRESIMS *m*/*z* 517.2770 [M + Na]^+^ (calcd. for C_27_H_42_O_8_Na, 517.2772, Δ + 0.4 ppm).

Virescenoside Z_13_ (**5**)*:* amorphous solids; [*α*]${}_{D}^{20}$ −69.2 (*c* 0.13, MeOH); ^1^H and ^13^C NMR data, see Table 1 and Table 2, Appendix A; HRESIMS *m*/*z* 533.2718 [M + Na]^+^ (calcd. for C_27_H_42_O_9_Na, 533.2721, Δ + 0.6 ppm).

Virescenoside Z_14_ (**6**): amorphous solids; [*α*]${}_{D}^{20}$ −44.0 (*c* 0.10, MeOH); UV (MeOH) *λ*_max_ (log *ε*) 249 (3.81) nm; ^1^H and ^13^C NMR data, see Table 1 and Table 3, Appendix A; HRESIMS *m*/*z* 547.2508 [M + Na]^+^ (calcd. for C_27_H_40_O_10_Na, 547.2514, Δ +1.0 ppm).

Virescenoside Z_15_ (**7**): amorphous solids; [*α*]${}_{D}^{20}$ + 17.3 (*c* 0.15, MeOH); UV (MeOH) *λ*_max_ (log *ε*) 248 (3.99) nm; ^1^H and ^13^C NMR data, see Table 1 and Table 3, Appendix A; HRESIMS *m*/*z* 547.2515 [M + Na]^+^ (calcd. for C_27_H_40_O_10_Na, 547.2514, Δ −0.2 ppm).

Virescenoside Z_16_ (**8**): amorphous solids; [*α*]${}_{D}^{20}$ −78.0 (*c* 0.05, MeOH); ^1^H and ^13^C NMR data, see Table 1 and Table 3, Appendix A; HRESIMS *m*/*z* 515.2617 [M + Na]^+^ (calcd. for C_27_H_40_O_8_Na, 515.2615, Δ−0.4 ppm).

Virescenoside Z_17_ (**9**): amorphous solids; [*α*]${}_{D}^{20}$ −60.0 (*c* 0.10, MeOH); ^1^H and ^13^C NMR data, see Table 1 and Table 3, Appendix A; HRESIMS *m*/*z* 575.3194 [M + Na]^+^ (calcd. for C_29_H_48_O_9_Na, 575.3191, Δ‒0.5 ppm), *m/z* 551.3229 [M‒H]^−^ calcd. for C_29_H_47_O_9_, 551.3226, Δ ‒0.6 ppm).

Virescenoside Z_18_ (**10**): amorphous solids; [*α*]${}_{D}^{20}$ ‒32.5 (*c* 0.12, MeOH); ^1^H and ^13^C NMR data, see Table 1 and Table 3, Appendix A; HRESIMS *m*/*z* 517.2773 [M + Na]^+^ (calcd. for C_27_H_42_O_8_Na, 517.2772, Δ‒0.3 ppm).

Virescenoside F (**11**): amorphous solids; ^1^H NMR (700 MHz, CD_3_OD) δ: 5.80 (1H, dd, J = 10.8, 17.4 Hz, H-15), 5.38 (1H, m, H-7), 4.92 (1H, dd, J = 1.4, 17.4 Hz, H-16b), 4.86 (1H, d, J = 3.0 Hz, H-1′), 4.85 (1H, dd, J = 1.4, 10.8 Hz, H-16a), 4.27 (1H, t, J = 3.7 Hz, H-4′),4.24 (1H, d, J = 4.3 Hz, H-5′), 4.12 (1H, d, J = 9.8 Hz, H-19b), 3.93 (1H, dd, J = 3.3, 8.3 Hz, H-3′), 3.78 (1H, dd, J = 3.0, 8.0 Hz, H-2′), 3.76 (1H, m, H-2), 3.73 (1H, d, J = 9.8 Hz, H-19a), 2.98 (1H, d, J = 9.8 Hz, H-3), 2.11 (1H, dd, J = 4.2, 12.5 Hz, H-1β), 2.04 (1H, m, H2-6), 1.99 (1H, m, H-14α), 1.92 (1H, dd, J = 2.8, 14.4 Hz, H-14β), 1.73 (1H, m, H-9), 1.60 (1H, m, H-11α), 1.50 (1H, m, H-12α), 1.39 (1H, m, H-12β), 1.39 (1H, m, H-11β), 1.34 (1H, dd, J = 5.8, 10.7 Hz, H-5), 1.14 (3H, s, Me-18), 1.11 (1H, d, J = 12.3 Hz, H-1α), 0.92 (3H, s, Me-20), 0.86 (3H, s, Me-17). ^13^C NMR (176 MHz, CD_3_OD) δ: 174.0 (C-6′), 151.9 (C-15), 136.9 (C-8), 123.2 (C-7), 110.4 (C-16), 103.5 (C-1′), 85.8 (C-3), 76.6 (C-5′), 74.3 (C-19), 70.9 (C-2′), 70.4 (C-3′), 70.2 (C-4′), 69.7 (C-2), 54.1 (C-9), 53.4 (C-5), 48.0 (C-1), 47.6 (C-14), 44.4 (C-4), 38.4 (C-13), 37.9 (C-10), 37.9 (C-12), 25.1 (C-6), 24.2 (C-18), 22.6 (C-17), 22.1 (C-11), 17.7 (C-20); Appendix A; HRESIMS m/z 519.2563 [M + Na]^+^ (calcd. for C_26_H_40_O_9_Na, 519.2565, Δ + 0.3 ppm).

Virescenoside G (**12**): amorphous solids; ^1^H NMR (700 MHz, CD_3_OD) δ: 5.80 (1H, dd, J = 10.8, 17.5 Hz, H-15), 5.37 (1H, m, H-7), 4.92 (1H, dd, J = 1.6, 17.5 Hz, H-16b), 4.86 (1H, d, J = 3.3 Hz, H-1′), 4.84 (1H, dd, J = 1.6, 10.8 Hz, H-16a), 4.30 (1H, t, J = 3.5 Hz, H-4′),4.24 (1H, d, J = 3.7 Hz, H-5′), 4.10 (1H, d, J = 9.9 Hz, H-19b), 3.93 (1H, dd, J = 3.3, 8.7 Hz, H-3′), 3.84 (1H, d, J = 9.9 Hz, H-19a), 3.78 (1H, dd, J = 3.3, 8.7 Hz, H-2′), 3.24 (1H, dd, J = 4.0, 11.8 Hz, H-3), 2.07 (1H, m, H-6α), 2.01 (1H, m, H-6β), 1.97 (1H, m, H-14α), 1.91 (1H, dd, J = 2.7, 14.0 Hz, H-14β), 1.90 (1H, dd, J = 3.4, 13.4 Hz, H-1β), 1.74 (1H, dd, J = 3.5, 11.8 Hz, H-2β),1.68 (1H, dd, J = 3.5, 7.5 Hz, H-2α),1.66 (1H, dd, J = 3.7, 7.7 Hz, H-9), 1.57 (1H, m, H-11α), 1.47 (1H, td, J = 2.9, 9.1 Hz, H-12β), 1.37 (1H, m, H-12α), 1.38 (1H, m, H-11β), 1.26 (1H, dd, J = 4.5, 12.1 Hz, H-5), 1.23 (1H, m, H-1α), 1.12 (3H, s, Me-18), 0.86 (3H, s, Me-17), 0.85 (3H, s, Me-20). ^13^C NMR (176 MHz, CD_3_OD) δ: 173.7 (C-6′), 152.0 (C-15), 137.1 (C-8), 123.2 (C-7), 110.4 (C-16), 103.9 (C-1′), 81.7 (C-3), 76.7 (C-5′), 73.9 (C-19), 70.6 (C-4′), 70.4 (C-3′), 70.1 (C-2′), 54.1 (C-9), 53.7 (C-5), 47.7 (C-14), 43.7 (C-4), 40.0 (C-1), 38.4 (C-13), 37.9 (C-12), 36.8 (C-10), 29.4 (C-2), 25.0 (C-6), 23.6 (C-18), 22.6 (C-17), 22.1 (C-11), 17.0 (C-20); Appendix A; HRESIMS m/z 503.2617 [M + Na]^+^ (calcd. for C_26_H_40_O_8_Na, 503.2615, Δ−0.3 ppm).

Lactone of virescenoside G (**13**): amorphous solids; ^1^H NMR (700 MHz, DMSO-d_6_) *δ*: 5.80 (1H, dd, *J* = 10.8, 17.6 Hz, H-15), 5.66 (1H, d, *J* = 7.8 Hz, 5′-OH), 5.56 (1H, d, *J* = 5.6 Hz, 2′-OH), 5.37 (1H, m, H-7), 5.29 (1H, d, *J* = 3.4 Hz, 2′-OH), 4.93 (1H, dd, *J* = 1.7, 17.6 Hz, H-16b), 4.85 (1H, dd, *J* = 1.7, 10.7 Hz, H-16a), 4.68 (1H, d, *J* = 6,8 Hz, H-1′), 4.41 (1H, dd, *J* = 5.6, 7.8 Hz, H-5′), 4.39 (1H, brs, H-3′), 4.14 (1H, d, *J* = 11.0 Hz, H-19b), 4.12 (1H, dd, *J* = 3.4, 5.8 Hz, H-4′), 3.55 (1H, ddd, *J* = 1.5, 5.4, 6.8 Hz, H-2′), 3.49 (1H, dd, *J* = 4.0, 11.9 Hz, H-3), 3.48 (1H, d, *J* = 11.0 Hz, H-19a), 2.22 (1H, dd, *J* = 3.1, 12.4 Hz, H-2b), 1.93 (1H, m, H-6β), 1.91 (1H, m, H-14α), 1.88 (1H, m, H-1β), 1.87 (1H, m, H-14β), 1.71 (1H, m, H-6α), 1.64 (1H, m, H-9), 1.55 (1H, m, H-11b), 1.46 (1H, m, H-2a), 1.44 (1H, m, H-12b), 1.31 (2H, m, H-11a, H-12a), 1.25 (3H, s, Me-18), 1.22 (1H, dd, *J* = 2.5, 9.3 Hz, H-5), 1.12 (1H, dt, *J* = 3.1, 12.5 Hz, H-1α), 0.89 (3H, s, Me-20), 0.82 (3H, s, Me-17). ^13^C NMR (176 MHz, DMSO-d_6_) *δ*: 174.0 (C-6′), 149.9 (C-15), 135.5 (C-8), 121.1 (C-7), 109.8 (C-16), 93.8 (C-1′), 84.0 (C-3′), 80.1 (C-3), 71.4 (C-2′), 70.1 (C-4′), 68.5 (C-5′), 68.4 (C-19), 50.8 (C-9), 49.9 (C-5), 45.4 (C-14), 36.5 (C-13), 36.1 (C-4), 35.9 (C-1), 35.5 (C-12), 34.7 (C-10), 25.5 (C-18), 21.7 (C-6), 21.3 (C-17), 21.2 (C-2), 19.7 (C-11), 15.7 (C-20); Appendix A; HRESIMS *m*/*z* 485.2508 [M + Na]^+^ (calcd. for C_26_H_38_O_7_Na, 485.2510, Δ + 0.4 ppm).

Aglycon of virescenoside A (**14**): amorphous solids; ^1^H NMR (500MHz, CD_3_OD) δ: 5.80 (1H, dd, J = 10.8, 17.5 Hz, H-15), 5.37 (1H, brs, H-7), 4.93 (1H, dd, J = 1.4, 17.5 Hz, H-16b), 4.85 (1H, dd, J = 1.4, 10.8 Hz, H-16a), 4.14 (1H, d, J = 11.2 Hz, H-19b), 3.79 (1H, ddd, J = 4.3, 9.8, 11.7 Hz, H-2), 3.50 (1H, d, J = 11.2 Hz, H-19a), 3.09 (1H, d, J = 9.8 Hz, H-3), 2.11 (1H, dd, J = 4.3, 12.6 Hz, H-1β), 1.98 (1H, m, H-6β), 1.97 (1H, m, H-14α), 1.91 (1H, dd, J = 2.2, 13.7 Hz, H-14β), 1.92 (1H, m, H-6α), 1.73 (1H, m, H-9), 1.61 (1H, dt, J = 3.9, 10.0 Hz, H-11β), 1.50 (1H, d, J = 8.7 Hz, H-12α), 1.39 (2H, m, H-11α, H-12β), 1.35 (1H, dd, J = 4.2, 12.0 Hz, H-5), 1.21 (3H, s, Me-18), 1.12 (1H, t, J = 12.3 Hz, H-1α), 0.93 (3H, s, Me-20), 0.86 (3H, s, Me-17). ^13^C NMR (125 MHz, CD_3_OD) δ: 151.9 (C-15), 137.1 (C-8), 123.1 (C-7), 110.4 (C-16), 86.5 (C-3), 69.7 (C-2), 66.6 (C-19), 54.0 (C-9), 53.0 (C-5), 47.7 (C-1), 47.6 (C-14), 44.4 (C-4), 38.4 (C-13), 37.9 (C-10), 37.8 (C-12), 24.8 (C-6), 24.5 (C-18), 22.6 (C-17), 22.1 (C-11), 17.9 (C-20), Appendix A; HRESIMS *m*/*z* 343.2241 [M + Na]^+^ (calcd. for C_20_H_32_O_3_Na, 343.2244, Δ + 0.8 ppm).

### 3.5. Urease Inhibition Assay

The reaction mixture consisting of 25 µL enzyme solution (urease from *Canavalia ensiformis*, Sigma, 1U final concentration) and 5 µL of test compounds dissolved in water (10–300.0 µM final concentration) was preincubated at 37 °C for 60 min in 96-well plates. Then 55 µL of phosphate buffer solution with 100 µM urea was added to each well and incubated at 37 °C for 10 min. The urease inhibitory activity was estimated by determining of ammonia production using indophenol method [26]. Briefly, 45 µL of phenol reagent (1% w/v phenol and 0.005% w/v sodium nitroprusside) and 70 µL of alkali reagent (0.5% w/v NaOH and 0.1% active chloride NaOCl) were added to each well. The absorbance was measured after 50 min at 630 nm using a microplate reader Multiskan FC (Thermo Scientific, Canada). All the reactions were performed in triplicate in a final volume of 200 µL. The pH was maintained 7.3–7.5 in all assays. DMSO 5% was used as a positive control.

### 3.6. Reactive Oxygen Species (ROS) Level Analysis in LPS-Treated Cells

The suspension of macrophages on 96-well plates (2 × 104 cells/well) were washedwith the PBS and treated with 180 µL/well of the tested compounds (10 μM) for 1 h and 20 µL/well LPS from E. coli serotype 055:B5 (Sigma, 1.0 μg/mL), which were both dissolved in PBS and cultured at 37 °C in a CO2-incubator for one hour. For the ROS levels measurement, 200 μL of 2,7-dichlorodihydrofluorescein diacetate (DCF-DA, Sigma, final concentration 10 μM) fresh solution was added to each well, and the plates were incubated for 30 min at 37 °C. The intensity of DCF-DA fluorescence was measured at λex 485 n/λem 518 nm using the plate reader PHERAstar FS (BMG Labtech, Offenburg, Germany) [27].

### 3.7. Reactive Nitrogen Species (RNS) Level Analysis in LPS-Treated Cells

The suspension of macrophages on 96-well plates (2 × 104 cells/well) were washed withthe PBS and treated with 180 µL/well of the tested compounds (10 μM) for 1 h and 20 µL/well LPS from E. coli serotype 055:B5 (Sigma, 1.0 μg/mL), which were both dissolved in PBS and cultured at 37 °C in a CO2-incubator for one hour. For the RNS levels measurement, 200 μL Diaminofluorescein-FM diacetate (DAF FM-DA, Sigma, final concentration 10 μM) fresh solution was added to each well, and the plates were incubated for 40 min at 37 °C, then replaced with fresh PBS, and then incubated for an additional 30 min to allow complete de-esterification of the intracellular diacetates. The intensity of DAF FM-DA fluorescence was measured at λex 485 n/λem 520 nm using the plate reader PHERAstar FS (BMG Labtech, Offenburg, Germany).

### 3.8. Peritoneal Macrophage Isolation

Mice BALB/c were sacrificed by cervical dislocation. Peritoneal macrophages were isolated using standard procedures. For this purpose, 3 mL of PBS (pH 7.4) was injected into the peritoneal cavity and the body intensively palpated for 1–2 min. Then the peritoneal fluid was aspirated with a syringe. Mouse peritoneal macrophage suspension was applied to a 96-well plate left at 37 °C in an incubator for 2 h to facilitate attachment of peritoneal macrophages to the plate. Then a cell monolayer was triply flushed with PBS (pH 7.4) for deleting attendant lymphocytes, fibroblasts and erythrocytes and cells were used for further analysis.

All animal experiments were conducted in compliance with all rules and international recommendations of the European Convention for the Protection of Vertebrate Animals used for experimental and other scientific purposes. All procedures were approved by the Animal Ethics Committee at the G. B. Elyakov Pacific Institute of Bioorganic Chemistry, Far Eastern Branch of the Russian Academy of Sciences (Vladivostok, Russia), according to the Laboratory Animal Welfare guidelines.

### 3.9. Statistical Analysis

Average value, standard error, standard deviation and p-values in all experiments were calculated and plotted on the chart using SigmaPlot 3.02 (Jandel Scientific, San Rafael, CA, USA). Statistical difference was evaluated by t-test, and results were considered as statistically significant at p < 0.05.

## 4. Conclusions

Ten new diterpene glycosides, virescenosides Z_9_‒Z_18_ (**1**‒**10**) were isolated from a marine strain of *Acremonium striatisporum* KMM 4401 associated with the holothurian *Eupentacta fraudatrix*. Virescenoside Z_9_ (**1**) is an altroside of a new 7-oxo-isopimara-15-en-2α,3β,6α,8β-tetraol aglycon. Virescenosides Z_12_-Z_16_ (**4**‒**8**) were determined as the monosides having unique methyl esters of altruronic acid as their sugar moieties. Carbohydrate chain of virescenoside Z_18_ (**10**) was structurally identified as the methyl ester of mannuronic acid. The effects of some isolated glycosides and aglycons **15**‒**18** on urease activity and regulation of ROS and NO production in macrophages stimulated with lipopolysaccharide (LPC) were evaluated.

## Figures and Tables

**Figure 1 marinedrugs-17-00616-f001:**
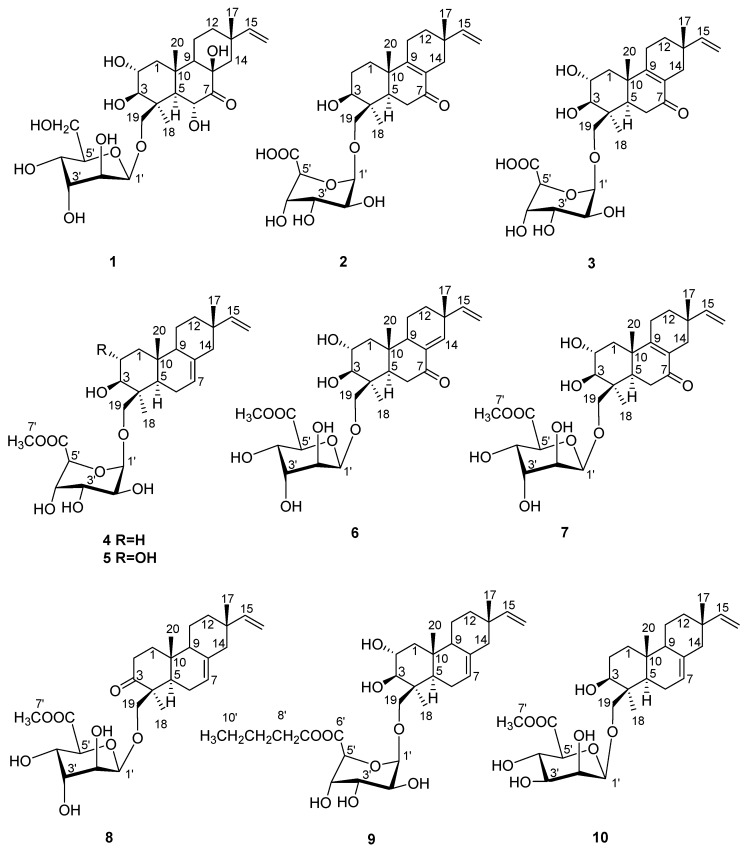
Chemical structures of **1**–**10**.

**Figure 2 marinedrugs-17-00616-f002:**
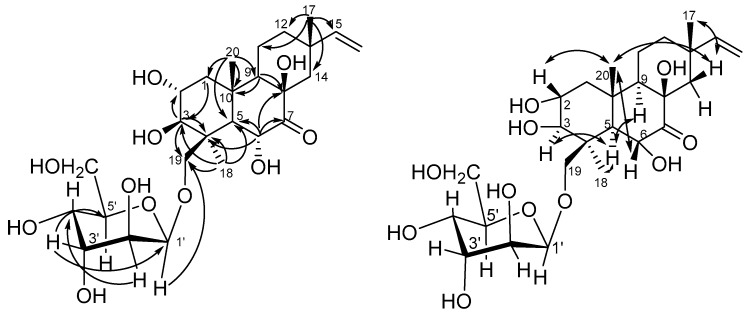
Key HMBC and NOESY correlations of **1**.

**Figure 3 marinedrugs-17-00616-f003:**
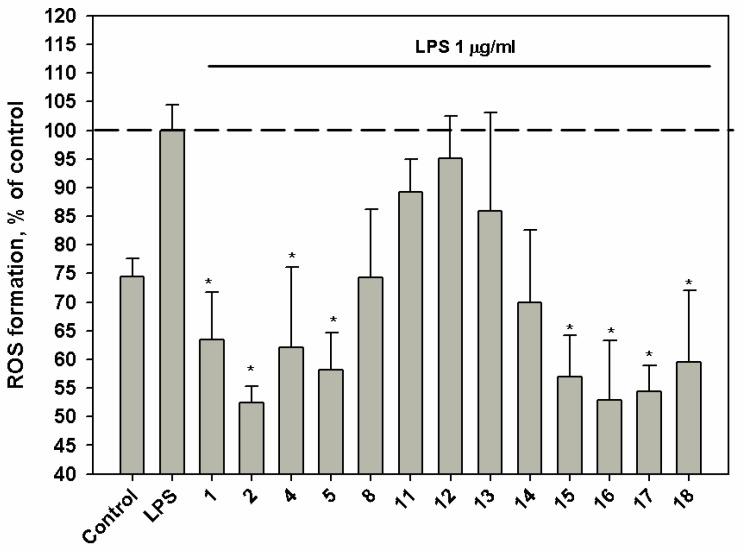
Influence of compounds upon ROS level in murine peritoneal macrophages, co-incubated with LPS from *E. coli*. The compounds were tested at a concentration of 10 *μ*M. Time of cell incubation with compounds was 1 h at 37 °C. * p < 0.05.

**Figure 4 marinedrugs-17-00616-f004:**
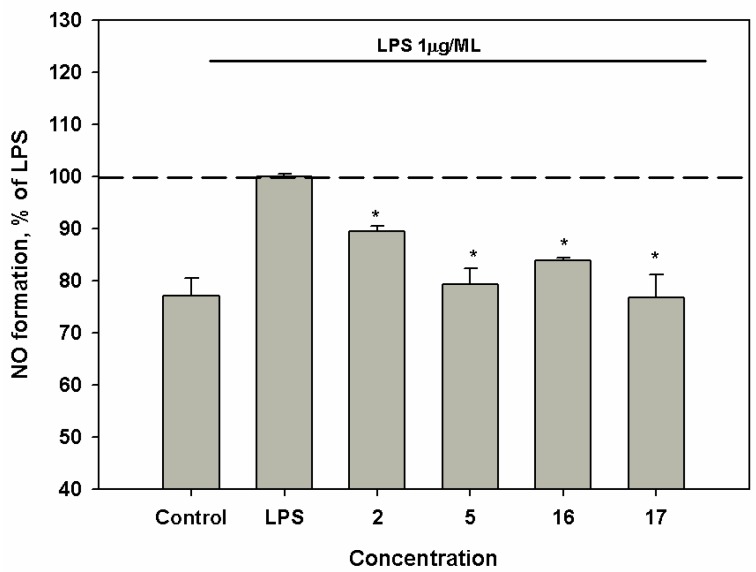
Influence of compounds upon RNS level in murine peritoneal macrophages, co-incubated with LPS from *E. coli*. The compounds were tested at a concentration of 1 *μ*M. Time of cell incubation with compounds was 1 h at 37 °C. * p < 0.05.

**Table 1 marinedrugs-17-00616-t001:** ^13^C NMR data (*δ* in ppm) for virescenosides Z_9_-Z_18_ (**1**–**10**).

Position	1^a^	2^b^	3^b^	4^c^	5^b^	6^c^	7^b^	8^c^	9^b^	10^d^
1	46.9, CH_2_	35.9, CH_2_	44.1, CH_2_	40.0, CH_2_	48.0, CH_2_	46.6, CH_2_	44.1, CH_2_	40.3, CH_2_	48.1, CH_2_	38.5, CH_2_
2	69.1, CH	29.0, CH_2_	69.6, CH	29.4, CH_2_	69.5, CH	69.3, CH	69.4, CH	37.3, CH_2_	69.5, CH	28.6, CH_2_
3	84.7, CH	80.0, CH	84.6, CH	81.5, CH	85.6, CH	84.7, CH	84.4, CH	218.4, C	85.6, CH	79.0, CH
4	43.8, C	41.3, C	44.7, C	43.8, C	44.6, C	44.9, C	44.7, C	54.1, C	44.6, C	43.0, C
5	55.9, CH	51.8, CH	51.7, CH	53.5, CH	53.2, CH	51.9 CH	51.5, CH	55.5, CH	53.2, CH	51.5 CH
6	57.3, CH	37.3, CH_2_	37.6, CH_2_	25.2, CH_2_	25.3, CH_2_	39.3, CH_2_	37.8, CH_2_	25.8, CH_2_	25.3, CH_2_	24.6, CH_2_
7	178.0, C	202.9, C	202.7, C	123.1, CH	123.2, CH	203.1, C	202.8, C	123.1, CH	123.2, CH	122.8, CH
8	80.2, C	130.3, C	130.3, C	137.2, C	137.1, C	136.8, C	130.2, C	137.7, C	136.9, C	134.9, C
9	60.5, CH	168.7, C	167.8, C	54.1, CH	54.1, CH	52.6, CH	167.9, C	53.1, CH	54.1, CH	52.3, CH
10	43.8, C	43.9, C	42.2, C	36.8, C	37.9, C	38.4, C	42.2, C	37.0, C	37.9, C	35.5, C
11	18.4, CH_2_	24.9, CH_2_	25.0, CH_2_	22.1, CH_2_	22.2, CH_2_	21.0, CH_2_	25.0, CH_2_	22.1, CH_2_	22.2, CH_2_	20.5, CH_2_
12	34.3, CH_2_	35.3, CH_2_	35.2, CH_2_	37.9, CH_2_	37.8, CH_2_	35.6, CH_2_	35.2, CH_2_	37.8, CH_2_	37.8, CH_2_	36.4, CH_2_
13	35.8, C	36.0, C	36.1, C	38.4, C	38.4, C	40.4, C	36.1, C	38.4, C	38.4, C	37.1, C
14	49.8, CH_2_	34.9, CH_2_	34.9, CH_2_	47.7, CH_2_	47.6, CH_2_	146.5, CH	34.9, CH_2_	47.6, CH_2_	47.6, CH_2_	46.3, CH_2_
15	151.0, CH	147.0, CH	147.2, CH	152.0, CH	151.9, CH	148.6, CH	146.9, CH	151.9, CH	151.9, CH	150.6, CH
16	108.4, CH_2_	112.7, CH_2_	112.7, CH_2_	110.4, CH_2_	110.4, CH_2_	112.9, CH_2_	112.7, CH_2_	110.5, CH_2_	110.4, CH_2_	109.6, CH_2_
17	28.5, CH_3_	29.2, CH_3_	29.1, CH_3_	22.6, CH_3_	22.6, CH_3_	26.8, CH_3_	29,2, CH_3_	22.6, CH_3_	22.6, CH_3_	21.7, CH_3_
18	25.8, CH_3_	22.8, CH_3_	23.7, CH_3_	23.9, CH_3_	24.6, CH_3_	23.6, CH_3_	23.7, CH_3_	22.0, CH_3_	24.6, CH_3_	23.9, CH_3_
19	74.0, CH_2_	73.3, CH_2_	73.8, CH_2_	73.9, CH_2_	74.1, CH_2_	73.1, CH_2_	73,6, CH_2_	75.1, CH_2_	73.9, CH_2_	72.1, CH_2_
20	17.7, CH_3_	18.7, CH_3_	19.7, CH_3_	16.9, CH_3_	17.6, CH_3_	16.1, CH_3_	19,7, CH_3_	16.7, CH_3_	17.6, CH_3_	15.7, CH_3_
1′	101.2, CH	103.5, CH	103.3, CH	103.7, CH	103.3, CH	102.8, CH	102.8, CH	101.9, CH	103.3, CH	103.5, CH
2′	72.7, CH	71.1, CH	71.3, CH	70.9, CH	71.2, CH	71.6 CH	71.7, CH	72.1, CH	71.3, CH	71.6 CH
3′	72.3, CH	70.8, CH	71.1, CH	70.6, CH	71.1, CH	71.8, CH	72.0, CH	72.0, CH	71.2, CH	75.1, CH
4′	67.2, CH	69.9, CH	69.7 CH	69.9, CH	69.6, CH	69.1, CH	69.0, CH	68.7, CH	69.7, CH	70.0, CH
5′	75.7, CH	76.5, CH	76.6, CH	76.6, CH	76.4, CH	76.4, CH	76.1, CH	76.0, CH	76.4, CH	77.8, CH
6′	64.0, CH_2_	174.0, C	174.0, C	172.7, C	172.9, C	173.2, C	172.8, C	172.6, C	172.9, C	170.7, C
7′				53.3, CH_3_	53.3, CH_3_	53.4 CH_3_	53.3, CH_3_	53.3, CH_3_	66.8, CH_2_	51.8 CH_3_
8′									32.3, CH_2_	
9′									20.7, CH_2_	
10′									14.7, CH_3_	

^a^ Chemical shifts were measured at 176.04 in Pyr-d_5_. ^b^ Chemical shifts were measured at 176.04 in CD_3_OD. ^c^ Chemical shifts were measured at 125.77 in CD_3_OD. ^d^ Chemical shifts were measured at 125.77 in Pyr-d_5_.

**Table 2 marinedrugs-17-00616-t002:** ^1^H NMR data (*δ* in ppm, *J* in Hz) for virescenosides Z_9_-Z_13_ (**1**–**5**).

Position	1^a^	2^b^	3^b^	4^c^	5^b^
1	α: 1.54 t (11.5) β: 2.34 dd (4.6, 12.2)	α: 1.35 m β: 1.94 m	α: 1.23 m β: 2.17 dd (4.5, 12.8)	α: 1.22 dt (4.6, 13.5) β: 1.90 dd (3.5, 13.5)	α: 1.11 β: 2.11 dd (4.2, 12.6)
2	4.28 ddd (4.5, 9.3, 11.5)	α: 1.82 dd (3.5, 11.9) β: 1.75 dd (4.0, 13.6)	3.82 m	α: 1.74 dd (3.4, 11.8) β: 1.65 dd (3.0, 13.4)	3.76 m
3	3.61 d (9.3)	3.26 dd (4.0, 11.9)	2.99 d (9.8)	3.24 dd (4.1, 11.8)	2.98 d (9.8)
5	2.41 d (13.2)	1. 67 dd (3.6, 14.4)	1. 76 dd (3.5, 14.7)	1. 26 t (8.2)	1.34 dd (3.9, 11.4)
6	3.70 d (13.2)	α: 2.54 dd (3.6, 18.0) β: 2.64 dd (14.4, 18.0)	α: 2.56 dd (3.3, 18.2) β: 2.70 dd (14.7, 18.2)	2.03 m	2.03 m
7				5.38 brs	5.39 brs
9	1.93 t (7.5)			1.66 dd (3.9, 7.8)	1.74 m
11	α: 1.38 m β: 1.69 m	α: 2.23 m β: 2.27 m	α: 2.26 m β: 2.32 m	α: 1.38 m β: 1.58 m	α: 1.41 m β: 1.61 m
12	α: 1.87 β: 1.36	α: 1.35 m β: 1.62 m	α: 1.64 m β: 1.36 m	α: 1.37 m β: 1.48 dt (2.7, 8.9)	α: 1.50 dd (2.8, 12.1) β: 1.39 td (2.8, 11.5)
14	α: 2.37 d (14.0) β: 1.48 d (14.0)	α: 2.30 d (17.5) β: 1.93 d (17.5)	α: 2.31 m β: 1.94 dt (2.6, 17.9)	α: 1.97 brd (14.1) β: 1.91 dd (2.6, 14.1)	α: 1.99 m β: 1.92 dd (2.6, 14.1)
15	6.64 dd (10.8, 17.6)	5.70 dd (10.6, 17.5)	5.70 dd (10.8, 17.5)	5.80 dd (10.7, 17.5)	5.81 dd (10.8, 17.6)
16	a: 4.85 dd (1.8, 10.8) b: 4.96 dd (1.8, 17.6)	a: 4.82 dd (1.4, 17.5) b: 4.93 dd (1.4, 10.6)	a: 4.82 dd (1.5, 17.6) b: 4.93 dd (1.5, 10.8)	a: 4.84 dd (1.3, 10.7) b: 4.92 dd (1.3, 17.5)	a: 4.85 dd (1.4, 10.8) b: 4.93 dd (1.4, 17.6)
17	0.95 s	1.00 s	1.01 s	0.86 s	0.86 s
18	1.81 s	1.13 s	1.16 s	1.10 s	1.11 s
19	a: 4.23 d (9.9) b: 4.98 d (9.9)	a: 3.73 d (10.2) b: 4.17 d (10.2)	a: 3.67 d (10.4) b: 4.14 d (10.4)	a: 3.83 d (10.2) b: 4.04 d (10.2)	a: 3.72 d (10.3) b: 4.03 d (10.3)
20	1.28 s	1.14 s	1.21 s	0.87 s	0.95 s
1	5.43 d (1.2)	4.84 d (2.9)	4.82 d (2.5)	4.85 d (2.5)	4.85 d (2.7)
2′	4.56 dd (1.2, 3.9)	3.77 dd (2.7, 7.8)	3.77 dd (2.5, 7.4)	3.77 dd (2.8, 7.9)	3.77 m
3′	4.74 t (3.7)	3.93 dd (2.9, 7.7)	3.94 dd (3.0, 7.4)	3.89 dd (3.0, 7.9)	3.90 dd (3.3, 7.3)
4′	4.50 m	4.23 t (4.8)	4.20 dd (3.0, 5.7)	4.26 dd (3.0, 4.9)	4.23 dd (3.3, 5.6)
5′	4.69 d (3.2, 12.2)	4.24 d (4.8)	4.22 d (5.7)	4.28 d (4.9)	4.28 d (5.6)
6′	a: 4.40 dd (6.5, 12.3) b: 4.51 m				
7′				3.78 s	3.78 s

^a^ Chemical shifts were measured at 700.13 in Pyr-d_5_. ^b^ Chemical shifts were measured at 700.13 in CD_3_OD. ^c^ Chemical shifts were measured at 500.13 in CD_3_OD.

**Table 3 marinedrugs-17-00616-t003:** ^1^H NMR data (*δ* in ppm, *J* in Hz) for virescenosides Z_14_-Z_18_ (**6**–**10**)

Position	6^c^	7^b^	8^c^	9^b^	10^d^
1	α: 1.22 m β: 2.07 dd (4.3, 12.7)	α: 1.23 m β: 2.18 dd (4.5, 12.8)	α: 1.50 m β: 2.19 m	α: 1.11 m β: 2.11 dd (4.2, 12.5)	α: 1.15 dt β: 1.78 brd (3.9, 13.1)
2	3.79 m	3.80 dd (9.8, 13.9)	α: 2.84 dt (5.4, 14.2) β: 2.23 m	3.76 m	α: 1.85 m β: 1.92 m
3	3.04 d (9.9)	3.00 d (9.8)		2.98 d (9.8)	3.55 dd (4.0, 11.9)
5	1.73 dd (5.0, 13.8)	1. 75 dd (3.4, 14.7)	1.63 dd (4.1, 12.3)	1.34 dd (4.5, 11.8)	1.27 m
6	α: 2.59 dd (5.0, 19.0)	α: 2.53 dd (3.4, 18.2) β: 2.80 dd (14.7, 18.2)	α: 2.04 m β: 2.11 m	α: 2.01 m β: 2.06 m	α: 2.06 m β: 2.40 m
7			5.41 brs	5.38 m	5.30 m
9	2.13 m		1.76 m	1.74 m	1.60 m
11	α: 1.79 m β: 1.54 m	α: 2.26 m β: 2.31 m	α: 1.64 m β: 1.47 m	α: 1.41 m β: 1.61 m	α: 1.46 m β: 1.32 m
12	α: 1.54 m β: 1.67 m	α: 1.64 m β: 1.36 m	α: 1.51 β: 1.44	α: 1.50 m β: 1.39 m	α: 1.32 m β: 1.45 m
14	6.68 t (2.1)	α: 2.32 m β: 1.95 d (17.8)	α: 2.00 m β: 1.94 d (2.6, 14.0)	α: 1.99 m β: 1.92 dd (2.6, 14.1)	α: 2.03 brd (14.0) β: 1.94 brd (14.0)
15	5.83 dd (10.7, 17.5)	5.71 dd (10.8, 17.5)	5.81 dd (10.7, 17.5)	5.81 dd (10.8, 17.4)	5.87 dd (10.6, 17.4)
16	5.00 m	a: 4.83 dd (1.2, 17.5) b: 4.93 dd (1.2, 10.8)	a: 4.86 dd (1.4, 10.7) b: 4.94 dd (1.4, 17.5)	a: 4.85 dd (1.4, 10.8) b: 4.93 dd (1.4, 17.4)	a: 4.95 d (10.6) b: 5.02 d (17.4)
17	1.12 s	1,01 s	0.89 s	0.86 s	0.90 s
18	1.13 s	1.15 s	1.11 s	1.11 s	1.41 s
19	a: 3.68 d (10.3) b: 4.09 d (10.3)	a: 3.65 d (10,4) b: 4.11 d (10,4)	a: 3.90 d (9.8) b: 3.96 d (9.8)	a: 3.71 d (10.5) b: 4.04 d (1053)	a: 4.26 d (10.3) b: 4.59 d (10.3)
20	0.95 s	1.22 s	1.17 s	0.95 s	0.93 s
1′	4.78 brs	4.79 d (2.0)	4.75 d (1.9)	4.84 d (2.1)	4.97 brs
2′	3.76 m	3.76 m	3.66 dd (1.9, 5.6)	3.77 dd (2.5, 7.4)	4.55 d (3.2)
3′	3.91 dd (3.3, 6.3)	3.92 dd (3.3, 6.0)	3.89 dd (3.0, 5.6)	3.92 dd (3.0, 7.4)	4.14 dd (3.3, 9.4)
4′	4.15 dd (3.3, 7.0)	4.14 dd (3.3, 7.3)	4.08 dd (3.2, 8.0)	4.22 m	4.87 t (9.3)
5′	4.24 d (7.0)	4.24 d (7.3)	4.23 d (8.0)	4.25 d (5.6)	4.40 d (9.3)
7′	3.76 s	3.76 s	3.77 s	a: 4.15 dt (6.6, 10.7) b: 4.19 m	3.64 s
8′				a,b: 1.68 m	
9′				a,b: 1.45 m	
10′				0.96 t (7.5)	

^a^ Chemical shifts were measured at 700.13 in Pyr-d_5_. ^b^ Chemical shifts were measured at 700.13 in CD_3_OD. ^c^ Chemical shifts were measured at 500.13 in CD_3_OD. ^d^ Chemical shifts were measured at 500.13 in Pyr-d_5_.

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
