# Peer review of "Virescenosides from the Holothurian-Associated Fungus Acremonium striatisporum Kmm 4401"

_marinedrugs, 2019, doi:10.3390/md17110616_

Round 1

Reviewer 1 Report

The manuscript describes isolation and structural characterization of virescenosides from the marine fungus Acremonium striatisporum. The subject should be interesting for the readers of the Journal. The assignment of structures to the newly isolated substances on grounds of NMR and MS data seems to be correct. The manuscript is well-structured, except the missing sub-section 3.2 (there is no section between 3.1 and 3.3). The language level is sufficient with only minor errors.

The authors assayed urease inhibitory activity of aglycons of two (known) virescenosides, though the activity was negligible. The aglycons (in most cases prepared by acid hydrolysis of virescenosides) and some of the glycosylated substances were also evaluated for their ROS and NO production in macrophages. In this case some of them showed interesting downregulation of ROS production. The downregulation of NO production was only weak.

Just a few suggestions to improve the spelling of the manuscript:

Line 29       biological/biologically

L183+188   virescenosides/virescenoside

L220/221    in result/as a result       

L225          carry/carries

L229          in the other pathogenesis - reconsider?

L275          amphiphylic/amphiphilic

L378          Themo/Thermo

L382+391  ...suspension of macrophages on 96-well plates (2 x 104 cells/well) were washed the PBS - was washed with the PBS? please reconsider

L396         replace/replaced... incubate/incubated

After these minor improvements (spell check) I can recommend the manuscript for publication.

Author Response

Dear Reviewer,

Thank you for consideration and careful review of our manuscript.

We took into account all your suggestions and made changes to the text of the manuscript.

“L382+391  ...suspension of macrophages on 96-well plates (2 x 104 cells/well) were washed the PBS - was washed with the PBS? please reconsider”  Really, peritoneal macrophages were washed with the PBS (phosphate buffered saline).

The numbering of sub-sections in Materials and Methods has been revised and corrected.

Reviewer 2 Report

Abstract needs to be revised. 

At the same time, glycosylated secondary metabolites of marine fungi are relatively rare. Among them ribofuranosides, containing as aglycon moieties anthraquinones [1–3], diphenyl ethers [4,5], isocoumarin [6] and naphthyl derivatives [7).... Re-write these two sentence as one. 

Second part of the introduction sounds like the methods and discussion. Needs to be revised. 

Author Response

Dear Reviewer,

Thank you for consideration and careful review of our manuscript.

We revised Abstract and second part of introduction.

We re-write 2 and 3 sentences as one.